# Phages-bacteria interactions underlying the dynamics of polyhydroxyalkanoate-producing mixed microbial cultures via meta-omics study

Jian Yao,[1,2] Yan Zeng,[1,2] Xia Hong,[3] Meng Wang,[4] Quan Zhang,[4] Yating Chen,[2,5] Min Gou,[1,2,6] Zi-Yuan Xia,[1,2] Yue-Qin Tang[1,2]

**ABSTRACT**  The dynamics of the structure of polyhydroxyalkanoate-producing mixed microbial cultures (PHA-MMCs) during enrichment and maintenance is an unsolved problem. The effect of phages has been proposed as a cause of dynamic changes in community structure, but evidence is lacking. To address this question, five PHA-MMCs were enriched, and biological samples were sampled temporally to study the interactions between phage and bacterial members by combining metagenomics and metatranscriptomics. A total of 963 metagenome-assembled genomes (MAGs) and 4,294 phage operational taxonomic units (pOTUs) were assembled from bulk metagenomic data. The dynamic changes in the structure of phage and bacterial communities were remarkably consistent. Structural equation modeling analysis showed that phages could infect and lyse dominant species to vacate ecological niches for other species, resulting in a community succession state in which dominant species alternated. Seven key auxiliary metabolic genes (AMGs), *phaC*, *fadJ*, *acs*, *ackA*, *phbB*, *acdAB,* and *fadD*, potentially contributing to PHA synthesis were identified from phage sequences. Importantly, these AMGs were transcribed, indicating that they were in an active expression state. The meta-analysis provides the first catalog of phages in PHA-MMCs and the AMGs they carry, as well as how they affect the dynamic changes in bacterial communities. This study provides a reference for subsequent studies on understanding and regulating the microbial community structure of open microbial systems.

**IMPORTANCE**  The synthesis of biodegradable plastic PHA from organic waste through mixed microbial cultures (PHA-MMCs), at extremely low cost, has the potential for expanded production. However, the dynamics of dominant species in PHA-MMCs are poorly understood. Our results demonstrate for the first time the impact of phages on the structure of bacterial communities in the PHA-MMCs. There are complex interactions between the PHA producers (e.g., *Azomonas*, *Paracoccus*, and *Thauera*) and phages (e.g., Casadabanvirus and unclassified Hendrixvirinae). Phage communities can regulate the activity and structure of bacterial communities. In addition, the AMGs related to PHA synthesis may hitchhike during phage-host infection cycles, enabling their dissemination across bacterial communities, and phages may act as a critical genetic reservoir for bacterial members, facilitating access to PHA synthesis-related functional traits. This study highlights the impact of phages on bacterial community structure, suggesting that phages have the potential to be used as a tool for better controlling the microbial community structure of PHA-MMCs.

**KEYWORDS**  polyhydroxyalkanoates (PHAs), mixed microbial cultures (MMCs), community assembly, microbial community structure dynamics, phages-bacteria interactions, phages-host dynamics, auxiliary metabolic genes (AMGs)

Address correspondence to Yue-Qin Tang, tangyq@scu.edu.cn.

The authors declare no conflict of interest.

See the funding table on p. 16.

Mixed microbial cultures (MMCs), which use complex microbial communities composed of multiple microorganisms as biocatalysts, have achieved significant achievements in pollutant removal (1) and waste utilization (2). Polyhydroxyalkanoates (PHAs) production by MMCs is a prime example. Due to outstanding biodegradability and material properties, PHAs have the potential to replace petroleum-based plastics in the face of increasing plastic pollution and fossil resource depletion (3). The high cost of PHA production with pure culture techniques has limited the widespread use of PHA materials (4). As a result, PHA production with MMC techniques that can utilize inexpensive waste biomass has been proposed and is more likely to facilitate commercial scale-up of PHA production due to its low production costs (4).

Currently, the process for the production of PHAs by MMCs using complex waste biomass has three main steps (5). The process includes the following: (i) acid production: producing short-chain carboxylic acids (SCCAs, such as acetate, propionate, butyrate, valerate, and lactate) that serve as substrates of PHAs production by anaerobic fermentation; (ii) MMC enrichment: enriching microbial communities with PHAs-synthesizing capacity from activated sludge by applying selective pressure; (iii) PHAs accumulation: the maximum PHAs accumulation of enriched MMCs using SCCAs. Of these, the second step is the most important in the process. Nowadays, aerobic dynamic feeding (ADF) is the main strategy to enrich PHA-MMCs (6). Alternating cycles of the feast (carbon rich) and famine (carbon poor) in the ADF provide selection pressure. Because PHA producers can accumulate intracellular PHAs as a backup carbon source during the feast phase, they are better able to gain a competitive advantage during the famine phase (7). However, the dominant species in enriched microbial communities continue to change even under stable maintenance conditions, and this species turnover cannot be fully explained by interspecific competition (7).

In recent years, researchers have begun to uncover the effects of phages on the structure and function of microbial communities in natural environments (e.g., deep-sea sediment [8] and soil [9]) and engineered systems (e.g., municipal wastewater treatment plants [10–12]). Virulent phages hijack the hosts to produce progeny and eventually lyse the host, and host cell debris fuels other microorganisms, which can directly affect the structure of microbial communities (13, 14). In addition, both virulent and temperate phages can manipulate host metabolism by reprogramming host metabolic pathways or expressing auxiliary metabolic genes (AMGs) to influence the function of microbial communities. For example, in the ocean, many AMGs are involved in photosynthesis (15), the pentose phosphate pathway (16), nitrogen metabolism (17), and sulfur metabolism (18). Given that PHA-MMCs are cultivated in open systems, the bacterial communities within may also be influenced by phages. In light of the above considerations, we hypothesize that phages constitute a factor affecting the structure of bacterial communities in PHA-MMCs and may carry auxiliary metabolic genes associated with PHA synthesis.

In this study, we set out to test this hypothesis in five PHA-MMCs enriched with different SCCAs as the sole carbon sources. A total of 30 biological samples over the time scale were collected for the metagenomics and metatranscriptomics analyses. Our study aimed to explore the following questions: (i) identification of phages in PHA-MMCs and characterization of their interactions with bacteria; (ii) whether and how phages affect the structure of the bacterial community; and (iii) whether phages encode PHA synthesis-related AMGs and whether these AMGs are transcriptionally active. A total of 4,294 pOTUs were recovered, and the structural equation modeling (SEM) was used to discuss the effects of phages on the activity of the bacterial microorganisms and the microbial community structure. Host prediction of APC and prediction of host-encoded antiphage defense systems were performed to explore the characteristics of phage-host interactions during enrichment and maintenance processes. AMGs involved in PHA synthesis in phages were identified, and the complementary metatranscriptome analysis was conducted to verify the active expression of AMGs, which supported the new discovery of active phage-mediated PHA biosynthesis function.

It is the first study to elucidate the effect of phages on the community structure and function of PHA-MMCs. The results guide subsequent studies to construct efficient and robust PHA-MMCs.

## RESULTS AND DISCUSSION

### Overview of the bacterial and phage communities in the PHA-MMCs

To investigate phage diversity during the enrichment and maintenance process of PHA-MMCs, as well as the effect of phages on the microbial structure, five PHA-MMCs were enriched with acetate, propionate, butyrate, valerate, and lactate as the sole carbon source, respectively, and continuously cultured for a period of 145 days (Fig. 1A). The PHA synthesizing capacity of the five PHA-MMCs was rapidly increased in the first 17 days (enrichment phase) and remained stable thereafter (maintenance phase). The butyrate-enriched PHA-MMCs had the highest PHA synthesizing capacity, followed by the acetate, lactate, and valerate-enriched PHA-MMCs, and the lowest being the propionate-enriched PHA-MMCs (Fig. 2A). The higher energy cost may be one reason for the low efficiency of PHA synthesis using propionate as the carbon source. Two acetyl-CoA and one NADPH need to be input to convert one molecule of propionate into one molecule of (R)-3-hydroxyvaleryl-CoA, which is much higher than the energy input of the butyrate or valerate conversion pathway (19).

To further understand the dynamics of the bacterial and phage community structure of PHA-MMCs and the interactions between them, 30 time-sequenced samples were collected and subjected to meta-omics sequencing and analysis during the enrichment and maintenance of MMCs (Fig. 1B and C). From these metagenomes, 963 metagenome-assembled genomes (MAGs) with completeness ≥70% and contamination ≤10% were recovered by *de novo* assembly and binning. After clustering by dRep (20), 711 unique MAGs were obtained, of which 112 were from Reactor A, 144 from Reactor B, 179 from Reactor C, 141 from Reactor D, and 135 from Reactor E. Among five reactors, the bacterial MAGs belong to Proteobacteria ($n = 69$, 86, 98, 93, and 83 in Reactors A, B, C, D, and E, respectively) and Bacteroidota ($n = 25$, 38, 46, 25, and 28 in Reactors A, B, C, and E, respectively) dominated in the bacterial communities (Fig. S1). The distribution of these MAGs in the different genera is shown in Fig. S2 to S6. In general, *Azoarcus*, *Paracoccus*, *Azomonas*, *Thauera,* and *Brevundimonas* were the dominant genera which were consistent with previous studies that adopted the same enrichment strategy using

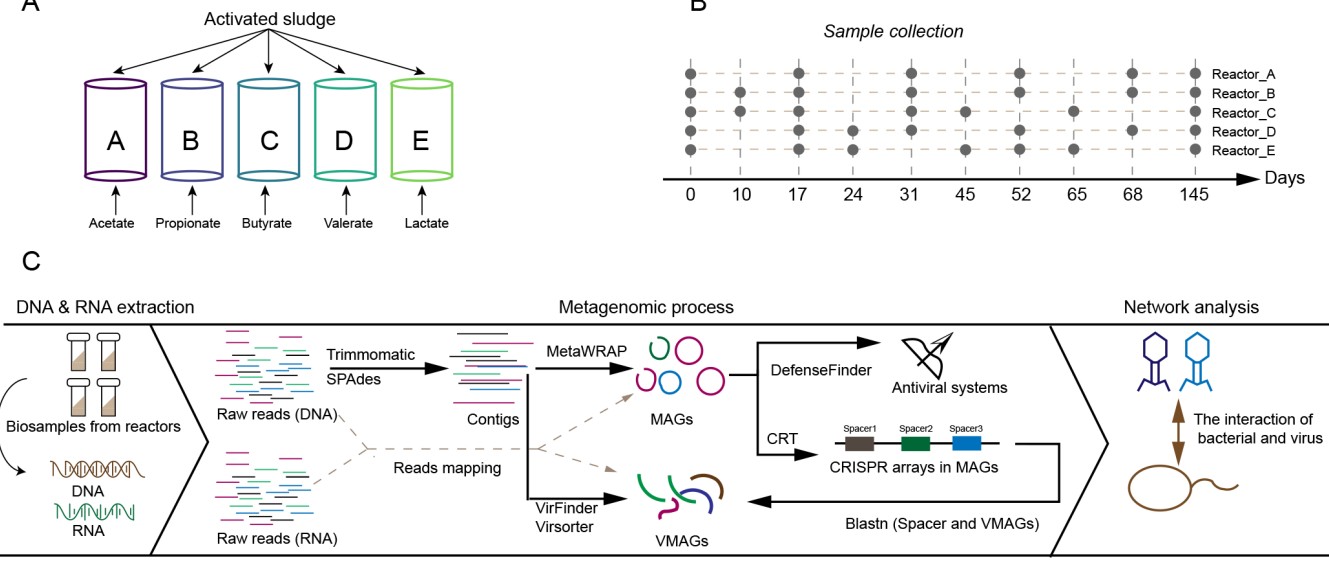

**FIG 1** Experimental design and analytical processes. (A) The five enriched polyhydroxyalkanoate-producing mixed microbial cultures (PHA-MMCs). (B) The distribution of biological samples collected. (C) The analysis processes of meta-omics data.

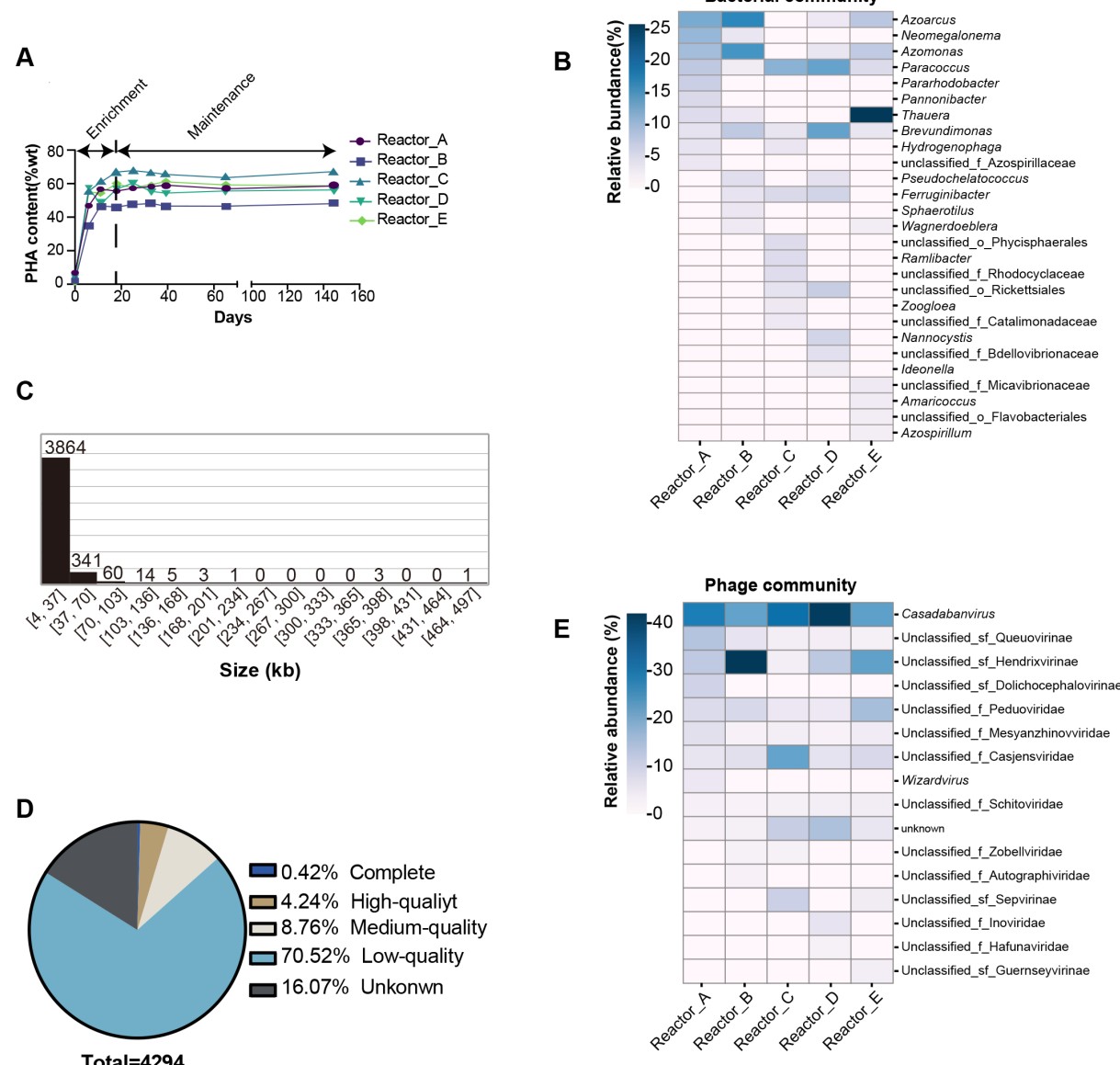

**FIG 2** The overview of bacterial and phage communities. (A) The maximum PHA accumulation content (wt%) of PHA-MMCs in each reactor. (B) The top 10 bacterial genera (relative abundance) in each reactor. (C) Distribution of pOTU sequence lengths. (D) The percentage of pOTUs of different qualities. (E) The top 10 pOTU genera (relative abundance) in each reactor.

activated sludge as inoculum seed (7, 21, 22) (Fig. 2B). However, the dominant genus was not the same in different reactors. For example, *Azoarcus* was dominant in Reactors A and B, rather than in Reactors C and D, and *Thauera* dominated in Reactor E, rather than in others, which might be due to the fact that different bacteria preferred different organic acids (21, 23–25).

The phage sequences were assembled and identified from bulk metagenomes containing sequences of diverse origins, which have been widely used to study viral communities in soil (26), deep-sea sediment (8), and deep-sea hydrothermal vents (27). In this study, VirSorter2 (28) and VirFinder (29) were used to identify assembled phage contigs (APCs) from bulk metagenomes, resulting in 5719 putative APCs (Contigs ≥ 5 kb (30)). These APCs were then clustered at 95% identity and 80% coverage to generate 4294 pOTUs. The size of these pOTUs ranged from 5,001 to 496,772 bp, in which six pOTUs were larger than 200 kb and possibly corresponded to giant phages (Fig. 2C). The

quality assessment of pOTUs was conducted by checkv (31), and 576 pOTUs (13.42%) were of medium quality and above, including 18 complete pOTUs (0.42%), high-quality pOTUs (4.24%) and medium-quality pOTUs (8.76%) (Fig. 2D). VirFinder and VirSorter2 are powerful tools for identifying phage sequences from metagenomes and have been used in many studies (32). The low proportion of high-quality phage sequences was also reported in previous studies, which may be due to insufficient understanding of the extremely complex virosphere (10, 30). The VirFinder uses the frequencies of DNA k-mers found in known viral genomes to train machine-learning classifiers to identify phage sequences (29). VirSorter2 uses domain percentages, gene content features, and key homology genes in a tree-based machine learning framework to classify phage reads (28). Compared with VirFinder, VirSorter2 is more advanced in that it moves beyond a single model to represent the virosphere for phage identification. However, VirFinder and VirSorter2 are heavily dependent on existing virus databases. The incompleteness of existing bacteriophage databases limits the output of better-quality phage identification results.

The life cycle of pOTUs was predicted based on the protein composition and associations by PhaTYP (33). Among these 4,294 pOTUs, 2,276 pOTUs were classified as temperate phages (53%) while 1,741 pOTUs were classified as virulent phages (40.5%). The average abundance of temperate phages identified was 1.7 times that of virulent phages. This may be due to the fact that temperate phages integrating into host genomes can be easily assembled from bulk metagenomic data (8). In addition, the lack of complete annotation of lytic-specific genes and the incomplete assembly of phage genomes also could lead to the underestimation of potent phage signals.

The taxonomy of pOTUs was classified by PhaGCN2, which is a GCN-based model and learns the species masking feature via a deep learning classifier (34). Totally, 2,007 pOTUs (46.7%) were classified at the phylum level, with the Uroviricota ($n = 1,902$) as the most dominant phylum in each reactor (Fig. S1). Zhang et al. studied the phages in activated sludge from three different sewage treatment plants, using three methods (PhaGCN2, CAT, and BLASTn to the IMG/VR database) for taxonomic classification, and the results also showed that Uroviricota was the most prevalent phylum (11). Some pOTUs could not be classified, due in part to the fact that the current viral phylogeny is still understudied, which also occurs in other studies (30, 35). Among the Uroviricota, 51 genera were identified for pOTUs, and the *Casadabanvirus* ($n = 740$, a genus in the class Caudoviricetes) was the most dominant genus of the class Caudoviricetes, followed by the three unclassified genera in Hendrixvirinae (subfamily), Casjensviridae (family), and Peduoviridae (family), respectively (Fig. 2E). This was consistent with the results of the study of Yuan et al. in which, phages in sludge from 32 wastewater treatment plants were taxonomically classified using vConTACT (10).

Consistent with the bacterial community, the abundance of members in the phage community changes dynamically over time (Tables S1 to S5). The mean rank shift (MRS) (36) was used to quantitatively characterize this dynamic change of phage and bacterial communities (Fig. S7). In short, the higher the MRS value is, the bigger the fluctuations in dominant taxa. In these communities, the MRS of phages at adjacent time points ranged from 400 to 1,000, which was much higher than that of bacterial members. This was partly because the number and diversity of phages far outstripped bacterial members in PHA-MMCs. In addition, there was a positive correlation between the MRS of phages and bacterial communities (Spearman's rho and *P*-value in Reactor A: 0.89 and 0.037; Reactor B: 0.65 and 0.1; Reactor C: 0.81 and 0.04; Reactor D: 0.87 and 0.02; Reactor E: 0.37 and 0.2). The positive correlation in Reactors B and E was not significant. This may be due to that MRS does not consider the loss and gain of members during community succession. In addition, the time-lagged association between changes in the number of phages and bacterial hosts may also contribute to this MRS insignificant positive correlation.

## Evidence that phages shape bacterial communities

To show the differences among the community structure of bacterial and phage communities in the different samples, the Bray-Curtis distance among samples was calculated and sorted by principal co-ordinates analysis (PCoA) (Fig. 3A and B). Sample points in each reactor did not cluster together during the late stages of maintenance (days 65, 68, and 145), suggesting that the bacterial community structure reached alternative states during maintenance, even when deterministic selection pressure was applied (alternating satiation and starvation phases). In addition, the structure of the phage communities in different samples also did not converge.

To investigate whether the dynamics of bacterial community structure could be mirrored in phage communities, these temporal changes in community structure were quantified by Bray-Curtis community dissimilarity values, and the Spearman's rho between bacterial and phage communities was calculated (Fig. 3C). The Bray-Curtis community dissimilarity values describe the degree of difference between

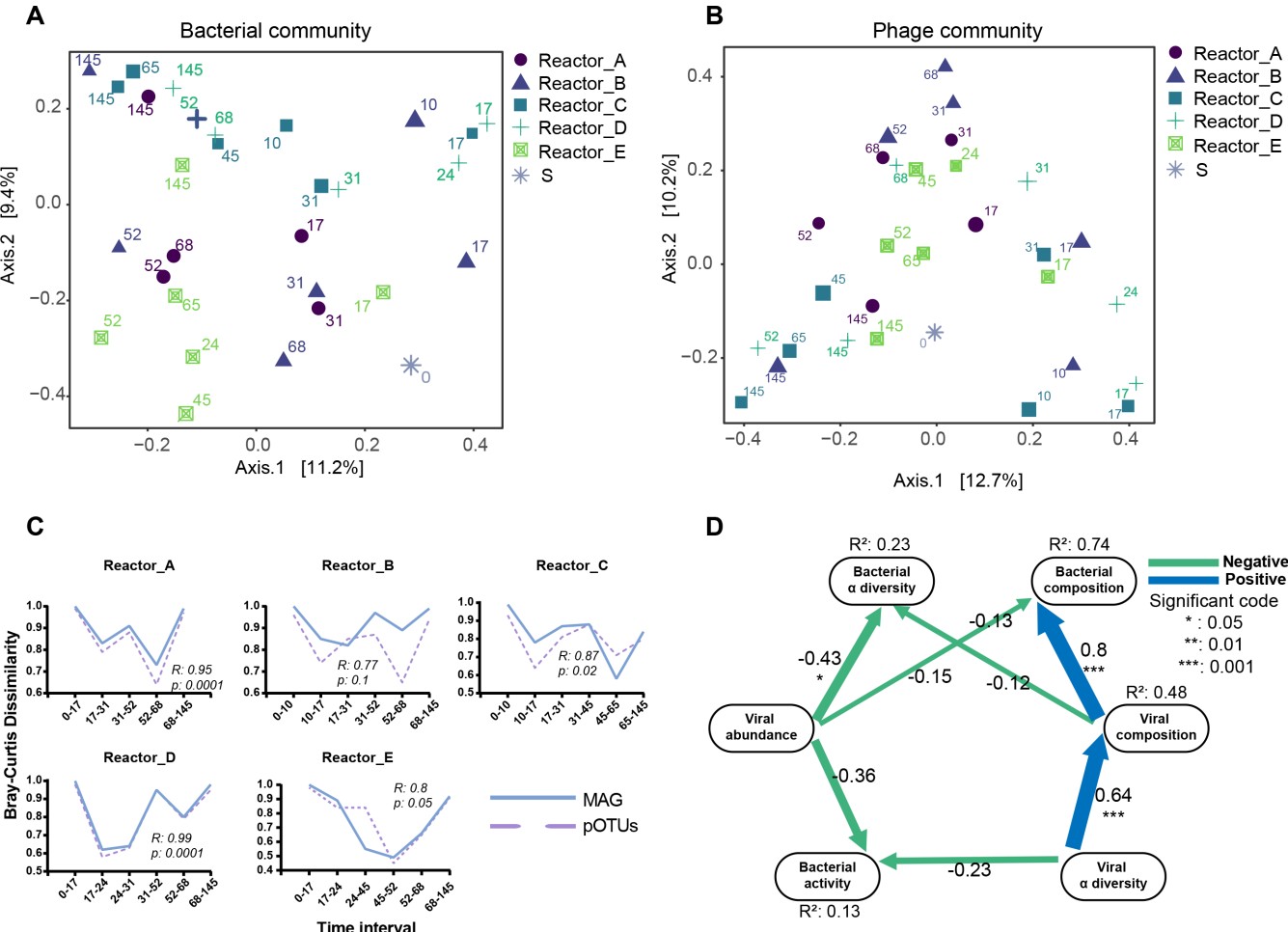

**FIG 3** Temporal dynamics of bacterial and phage communities The PCoA of bacterial communities (A) and phage communities (B). The "S" denotes activated sludge inoculum and the numbers indicate the days of sampling. (C) The Bray-Curtis community dissimilarity through days for bacterial (solid line) and phage (dashed line) communities. The dissimilarity illustrates the change in community composition from the previous time point. Spearman's correlation test was tested between the dissimilarity of bacterial and viral communities. (D) Path diagram of SEM showing the effect of phage communities on bacterial communities. Composition is represented by the PC1 from the Bray-Curtis dissimilarity-based principal coordinate analyses. The α diversity is represented by the Chao1 index. The phage abundance is the sum of the abundance value of each phage in the sample based on the metagenomic data. The bacterial activity is the sum of the activity value of each bacterial MAG in the sample based on the metatranscriptomic data. Numbers adjacent to the arrows are standardized path coefficients (*r*). "*" means the significant index (chi-squared test) $P \leq 0.05$ and "***" means the significant index $P \leq 0.001$. $R^2$ represents the proportion of variance explained for every dependent variable in the model.

the community structures of temporally adjacent samples; the larger the Bray-Curtis community dissimilarity value, the greater the difference. There was a significant positive relationship between the bacterial and phage communities shifts, and their dynamics mirrored well (Spearman's rho in Reactors A, B, C, D and E = 0.95, 0.77, 0.87, 0.99, and 0.8). These strong correlations may indicate that the bacterial and viral communities were constantly changing and that viral predation was occurring. The encounter ratio with the corresponding phage would be greatly increased by an increase in host abundance, which promotes the proliferation of phages (37, 38). As the virulent phages could hijack hosts for self-replication and then lyse the host cells, under phage predation, the number of hosts may gradually decrease, which negatively affects phage proliferation, leading to a subsequent decrease in the abundance of phages. However, more evidence is needed to prove phage predation on hosts, which is the key to showing that phage is one of the driving forces in changing the compositional structures of bacterial communities described above.

To solve this problem, the SEM was used to discern the causality and quantify the effects of the drivers (Fig. 3D). The significant positive correlation ($r$: 0.8, $P \leq 0.001$) between phage community composition and bacterial community composition was consistent with the above analyses (Fig. 3C), as phages were dependent on the host for survival. Importantly, there was a significant negative correlation ($r$: $-0.43$, $P \leq 0.05$) between phage abundance and the alpha diversity (Chao1) index of the bacterial community, in addition to a negative correlation ($r$: $-0.36$) with bacterial community activity. This might indicate that virulent phages hijacked and killed host cells during proliferation, thereby reducing the activity of the bacterial hosts. This further suggested that the synergistic changes between the phage and bacterial communities were not caused by the response of the temperate phages to the dynamics of hosts, as they benefit from the high growth rate of the host while limiting their lytic activity (37, 39).

This result might inspire us to develop a method to use phage to *in situ* regulate the microbial community structure of PHA-MMCs to improve their PHA synthesis efficiency. The efficiency of PHA synthesis by PHA-MMCs is generally lower than that by pure bacterial culture, partly because there are members in the microbial community that do not synthesize PHA, resulting in substrate waste (40). The selection pressure applied by the current enrichment strategy could not eliminate these non-PHA producers. If we could isolate and culture phages that can specifically lyse these non-PHA producers and add these phages into PHA-MMCs to specifically eliminate non-PHA producers, it is possible to obtain PHA-MMCs with higher PHA synthesis efficiency.

## Phage-host interaction dynamics

To specifically elucidate the interactions between phages and hosts, the putative phage-host linkages and the antiviral defense systems of hosts were screened. During the interaction between phages and bacteria, bacteria use the CRISPR-Cas system to add spacers to their genome to resist the next infection (41). This allows us to search for potential host-phage relationships by comparing spacer sequences in bacteria with pOTU sequences. Overall, a total of 16,725 spacers were recovered from all 30 metagenomes. All five reactors had a positive relationship between the number of spacers recovered and time (Fig. S8A). In addition, it was observed that the number of spacers also had a significant and strong positive relationship with the total number of pOTUs across all samples (Fig. S8B). These results suggested that the CRISPR-Cas system played an important role in the ongoing interaction between bacteria and phages and integrated matching spacers into CRISPR arrays through time.

Furthermore, the hosts of 200 high-quality pOTUs were searched using CRISPR-match and genomic homology matching. A total of 1,421 linkages were detected between 129 pOTUs (including 15 genera) and 409 hosts (including 73 genera). The linkages between phages and bacterial hosts were constructed as interaction networks at the genus level (Fig. 4A). This interaction network had 88 nodes and 198 links. The statistical characteristics of each node in the interaction network are found in Table S6.

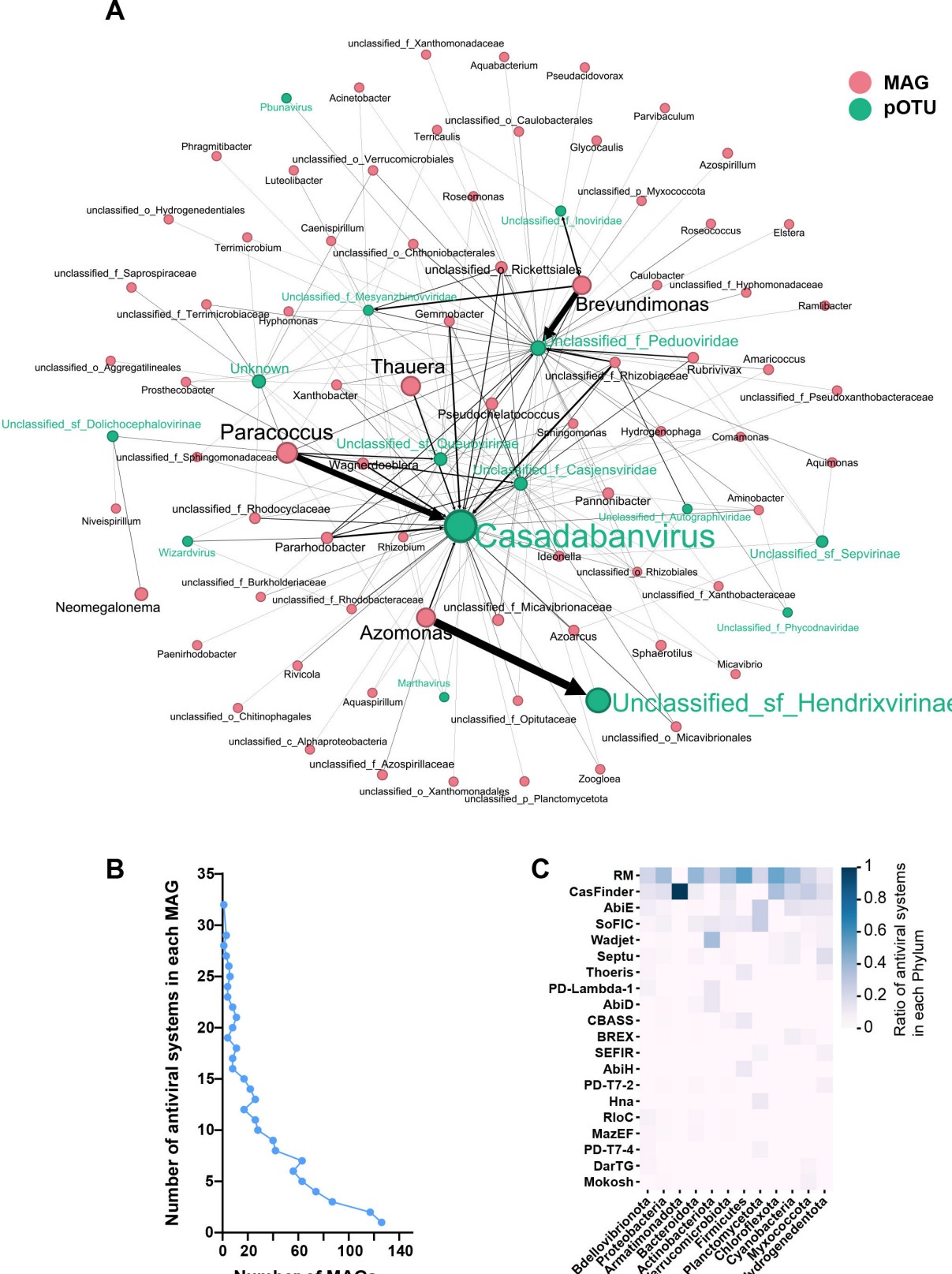

**FIG 4** The interaction between phages and hosts. (A) The linkages network of phages and hosts (all interactions between phages and hosts). The red nodes and green nodes represent the hosts and phages. The size of nodes represents the abundance of the genus. The size of the edges represents the number of linkages between phages and hosts. The "unclassified_f_Casjensviridae" represents the unclassified genus in the family Casjensviridae. The "Unclassified_sf_Sepvirinae" represents the unclassified genus in the subfamily "Sepvirinae." (B) The histogram of the number of antiviral systems in each host. (C) The proportion of the top 20 antiviral systems in each host phylum.

Among these nodes, 61 nodes had a degree greater than 1. This result might indicate that the phage-host linkage was not unique, but rather that a single phage had the potential to infect multiple hosts, or that a single host was infested by multiple phages, which was consistent with previous studies in oil wells (42), earthworm intestines (30), and acid mines (43). For example, the hosts of the phage genus *Casadabanvirus* were distributed among 47 bacterial genera, including PHA-synthesizing bacteria such as *Azoarcus*, *Hydrogenophaga*, and *Paracoccus*. Second, there tended to be a significant positive correlation between the abundance (of both phage and host) and the number of phage-host linkages at the genus level (Fig. S9). One reason for this might be that a high abundance of hosts that adapted to selective pressures (e.g., PHA producers) provided high encounter ratios for potential phages, and sufficient hosts also made these phages much more abundant than others (37). Another possible reason was that linkages between high-abundance phages and the host were more easily detected, as certain rare phages might not even be recovered and assembled with high quality. These dynamic linkages between phages and bacteria might have a potential role in maintaining community stability. These phages might selectively infect and lyse community members that were overabundant in the community, such as *Azoarcus* and *Paracoccus*, providing niches for other community members and preventing a single member from dominating the microbial community (37). This emergent state in which community members took turns to assume the leading position in the community had higher biodiversity and community resilience, leading to better community stability (44).

In the co-evolution of phage and host, the hosts are also able to adapt different antiviral systems to resist phage infection (45). DefenseFinder was used to discover the antiviral systems encoded in the bacterial MAGs and found that the antiviral arsenal of the prokaryotes was highly variable (Fig. 4B), which was consistent with Tesson's study (45). Overall, 86% of the MAGs encoded two or more types of antiviral weapons (Fig. 4B). However, as the number of encoded antiviral systems increased, the number of corresponding MAGs decreased rapidly, suggesting that encoding more antiviral systems was not a perfect survival strategy. Of the 109 antiviral systems detected, restriction-modification (RM) and CRISPR-Cas were dominant, accounting for 36.5% and 14.6% of the total, respectively (Fig. 4C). The RM and CRISPR-Cas provide defense through the degradation of viral nucleic acids, which appear to be the most widespread antiviral systems used by prokaryotes in a variety of habitats (30, 45, 46). The RM and CRISPR-Cas systems were the predominant antiviral systems in 10 of the 12 bacterial phyla, while Wadjet and SoFIC were the predominant antiviral systems in Actinobacteriota and Planctomycetota, respectively. In addition, hosts defended themselves against phages through various abortive infection systems, which were "altruistic" cell death systems that are activated by phage infection and limited viral replication (46). For example, some members of Cyanobacteria, Myxococcota, and Hydrogenedentota had the AbiE antiviral system which is encoded by bicistronic operons and functioned via a non-interacting (Type IV) bacteriostatic TA mechanism (47). Some members of Firmicutes had the Thoeris antiviral system. When triggered by phage infection, the ThsB sensor of the Thoeris defense systems produces signaling molecules, which activate the $NAD^+$ degrading activity of ThsA, leading to the depletion of the host $NAD^+$ pool and abortive infection (48). These results showed a fierce arms race between phages and bacteria. During this co-evolution process, bacteria can evolve a variety of antiviral defense systems. Diverse defense systems may help bacteria overcome or escape phages (49). In fact, the combined action of RM and CRISPR-Cas systems does enhance bacterial resistance to phages (50). This inspires us that if we could isolate phages infecting PHA producers and then strengthen their interaction with PHA producers, we may be able to cultivate PHA producers with stronger phage resistance. Adding these enhanced PHA producers into PHA-MMCs may enhance or stabilize the dominant position of PHA producers, thereby improving the overall PHA synthesis stability of the community.

To specifically characterize the temporal dynamics of the phage-host linkages in each reactor, we used network diagrams to show the phages associated with the host at

different time periods (Fig. S10 to S14). We calculated the sequence similarity among the corresponding phages of each host using VIRIDIC (51). The results showed that most of the phages had very low sequence similarity to each other's nucleotide sequence, suggesting that the bacterial members of PHA-MMCs could be infected by diverse phages (Table S7). The linkages of the host and phages changed dynamically at different time points. For example, in Reactor D (Fig. S13), the linkages between *Paracoccus* and phage 0111B24 were detected on days 52 and 145, but not on day 68. The loss of linkages or the appearance of new linkages also happened on the *Paracoccus* in Reactor E (Fig. S14) and the *Thauera* in Reactor A (Fig. S10). However, many "strong phages" that formed tight linkages with the host were detected at all times, such as the phage 1010E_19 with *Brevundimonas* in Reactor A (Fig. S10). Among these hosts, *Azomonas* (in Reactors B, C, and E) and *Brevundimonas* (in Reactors A and D) had a more solid linkage with the phage, even though they were in different reactors. While the phages infecting *Paracoccus* (in Reactors C, D, and E) and *Thaurea* (in Reactor A) were more variable at different time points. Furthermore, phage characteristics (e.g., lifestyle) were not significantly related to the diversity of these linkages. This indicated that the stability of host-phage linkages was more dependent on host characteristics.

## Potential impacts of AMGs on the metabolisms and the PHA synthesis of host

Phages can influence biogeochemical processes in the ocean (8), municipal wastewater treatment plants (10), and gut (30) by delivering AMGs while infecting hosts. In a recent study, Yuan et al. analyzed the phages in the sludge samples from 23 sewage treatment plants and found that these phages carried a large number of AMGs involved in carbon (*acpP* and *prsA*), nitrogen (*amoC*), sulfur (*cysH*), and phosphorus (*phoH*) metabolism, and these AMGs were active at the transcriptional level (10). In addition, researchers found that phages in paddy soil carried a large number of AMGs related to carbon fixation. *In situ* isotopic labeling experiments revealed that $^{13}CO_2$ emissions from the treatment with added lysogenic phage decreased by approximately 17.9% (52).

To investigate the effect of phages on the PHA anabolism of PHA-MMCs, AMGs encoded in pOTUs were predicted using DRAMv pipelines and functionally annotated by the KEGG database. These AMGs were generally involved in four types of activities, namely metabolism, environmental information processing, genetic informational processing, and cellular processes (Fig. S15). The high proportion of AMGs related to purine metabolism, pyrimidine metabolism, amino acid biosynthesis, mismatch repair, homologous recombination, and DNA replication might be due to their favorability for phage replication in the host, which was consistent with other studies (8, 10).

It was worth noting that some AMGs in fatty acid metabolism may be involved in the PHA biosynthetic pathway, converting SCCAs (acetate, propionate, butyrate, valerate, and lactate) into PHA, which needs further identification. Overall, SCCAs are converted into (R)-3-hydroxybutyryl-CoA and (R)-3-hydroxyvaleryl-CoA through a series of biochemical reactions, and then polymerized into poly(3-hydroxybutyrate) (PHB) and poly(3-hydroxyvalerate) (PHV) by PHA synthase. After the conversion of acetate and lactate to acetyl-CoA, two molecules of acetyl-CoA are condensed to acetoacetyl-CoA by beta-ketothiolase. The acetoacetyl-CoA is reduced to (R)-3-hydroxybutyryl-CoA by acetoacetyl-CoA reductase. Butyrate and valerate are converted to (R)-3-hydroxybutyryl-CoA and (R)-3-hydroxyvaleryl-CoA by the beta-oxidation pathway, respectively. As for propionate, it is first converted to propionyl-CoA. Together with acetyl-CoA, they serve as substrates for beta-ketothiolase, followed by the reduction of 3-ketovaleryl-CoA to (R)-3-hydroxyvaleryl-CoA.

Seven key AMGs involved in PHA synthesis were further screened and identified from the functionally annotated AMGs using the NCBI CD-search tool. These AMGs were distributed among 10 phages and 24 hosts (Fig. 5A). The *phaC* was the most important AMG, as it encodes a PHA polymerase that polymerizes the precursors (R)-3-hydroxybutanoyl-CoA and (R)-3-hydroxyvaleryl-CoA into poly(3-hydroxybutyrate) (PHB) and poly(3-hydroxyvalerate) (PHV) (53). The genes *fadJ* (3-hydroxybutanoyl-CoA

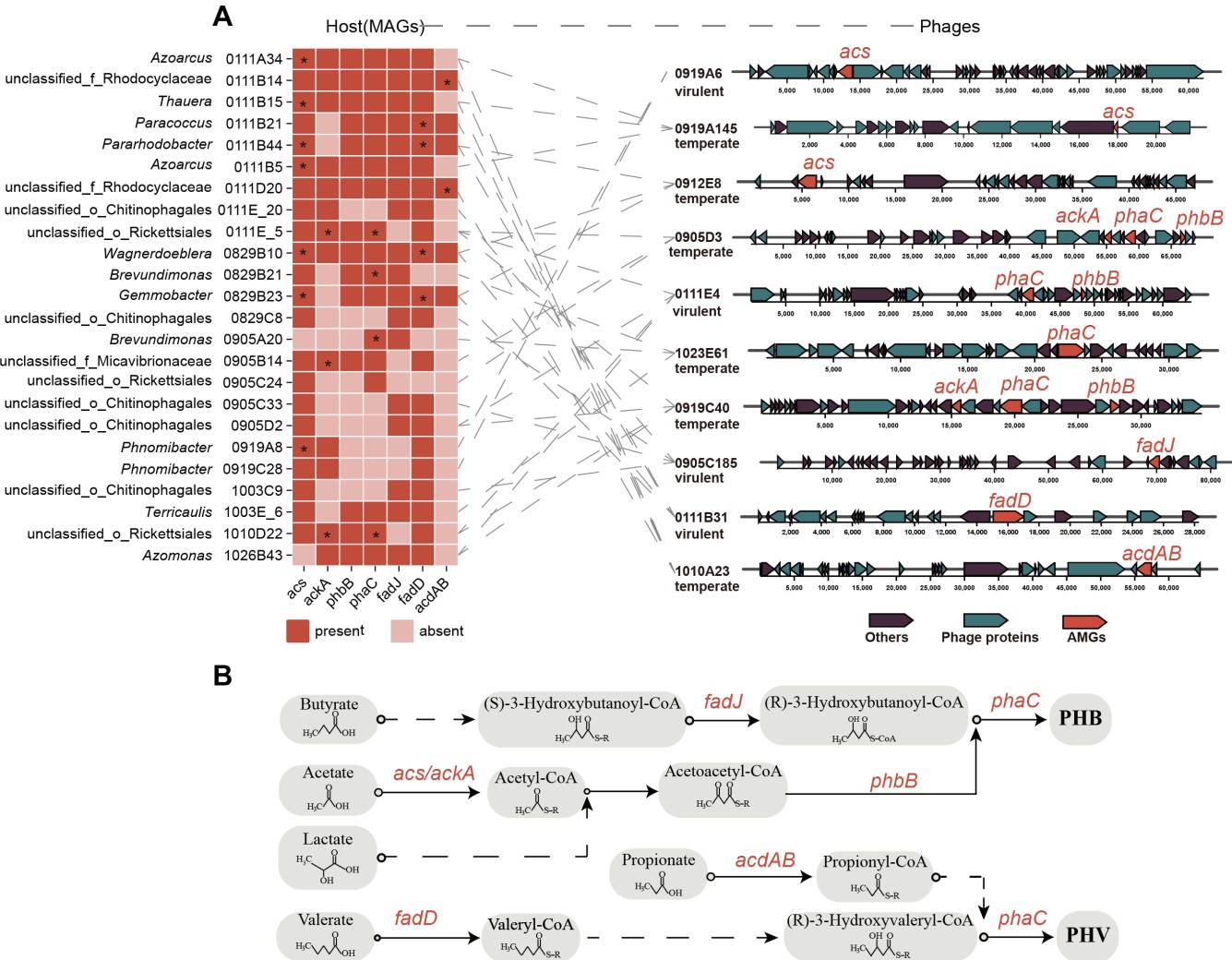

**FIG 5** The AMGs involved in PHA anabolism. (A) The distribution of AMGs in phages and hosts. The dotted line indicates the linkages between hosts and phages. The "*" indicates a high similarity (≥80%) between the host and phage AMGs. "Unclassified_f_X" represents the unclassified genus in family X. "Unclassified_o_X" represents the unclassified genus in order X. (B) Metabolic function of AMGs in PHA synthesis. The dotted lines represent the multi-metabolic pathways.

epimerase) (54) and *phbB* (acetoacetyl-CoA reductase) (55) catalyze the isomerization of (S)-3-hydroxybutanoyl-CoA and the reduction of acetoacetyl-CoA, respectively, to produce (R)-3-hydroxybutanoyl-CoA, providing precursors for PHA synthesis. In addition, *acs* (acetyl-CoA synthetase)/*ackA* (acetate kinase) (56, 57), *acdAB* (acetyl-CoA synthetase) (58), and *fadD* (long-chain fatty acid-CoA ligase) (59) catalyze the conversion of acetate, propionate, and valerate to acetyl-CoA, propionyl-CoA, and valeryl-CoA, respectively, facilitating the entry of organic acids into the PHA synthesis pathway (Fig. 5B).

Furthermore, the AMGs in phage and host were found to be diverse in sequence by phylogenetic analysis and sequence similarity calculation (Fig. S16 to S22). For example, the gene *phaC* encoding PHA synthase in hosts of a phage was not the same as AMG in this phage (Fig. S16). The *phaC* of phage 0905D3 was very similar to the nucleotide sequence in host YJ0111E.5, but significantly different from that in another host YJ0905C.24. In addition, *phaC* of 0905D3 had transcriptional activity in sample 0905C (Table S8). These phenomena were also present for the genes of *fadD*, *acdAB*, *acs*, and *ackA* (Fig. S17, S19, S21 and S22). Unexpectedly, in the case of both *phbB* and *fadJ*, the sequence similarity between the AMGs in the phage and host was less than 75% (Fig. S18 and S20). However, these AMGs carried by phages were transcriptionally active

in different samples (Table S8). These findings suggest that AMGs associated with PHA synthesis may hitchhike during phage-host infection cycles, enabling their dissemination across bacterial communities. Furthermore, phages may act as a critical genetic reservoir for bacterial members, facilitating access to PHA synthesis-related functional traits.

## Limitations and expectations of this study

In this study, we investigated the phage communities within PHA-MMCs and their potential impacts on bacterial community structure and function, though several methodological limitations remain, which we aim to address in future research. First, while cross-system comparisons among five PHA-MMCs enriched with distinct carbon sources were intended to identify generalized patterns of phage-mediated bacterial community dynamics, setting parallel reactor systems would further prove these findings. Subsequent studies should consider reactor replication and systematic validation across different systems to rigorously elucidate the role of phages in shaping bacterial communities.

The second limitation pertains to the prolonged intervals between sampling points, which compromised the ability to resolve more nuanced synchronous dynamics between phage and host abundance. Approximately half of the phages detected in this study exhibited multiple potential hosts, raising the question of whether such host generality may contribute to phage abundance and thereby obscure the coherence of phage and bacterial community structural shifts. Future studies should implement high-frequency sampling protocols within each of the PHA-MMC systems to rigorously track temporal fluctuations in phage and host abundance and concurrent community structural adaptations.

The third limitation concerns the need for further investigation into the influence of phage-encoded AMGs on bacterial community-mediated PHA synthesis. The finding that some phages within PHA-MMCs harbored transcriptionally active AMGs suggests that these phages may provide a critical genetic reservoir for bacterial members. However, the potential contribution of phage-carried AMGs on the PHA synthesis capacity of bacterial communities requires controlled experimental validation. Furthermore, whether lytic and lysogenic phages carrying AMGs exert equivalent impacts on the PHA synthesis efficiency of bacterial communities remains unresolved. Notably, excessive suppression of hosts by lytic phages could negate any potential enhancement of overall PHA yield. In subsequent studies, more rigorous controlled trials need to be conducted. It is needed to isolate lytic and lysogenic phages harboring AMGs, scale up their cultivation, supply them to PHA-MMCs, and systematically monitor changes in the PHA synthesis efficiency of bacterial communities.

## Conclusions

In this study, the dynamic interaction between phages and bacterial members in PHA-MMCs was investigated. In PHA-MMCs, phages could infect and lyse dominant species to vacate ecological niches for other species, resulting in a community succession state in which dominant species alternated. These results provide a potential explanation for the emergence of community members in open artificial microbial systems. In addition, while infecting different hosts, phages might express AMGs related to PHA synthesis to enhance the PHA synthesis capacity of community members. Of course, these interactions between bacteria and phages drawn from meta-omics analysis need to be further verified through rigorous validation experiments in the future. In subsequent studies, the *in situ* regulation of phages on PHA-MMCs can be considered to enhance its effectiveness in practical engineering applications. First, "phage therapy" could be used to add relevant phages to eliminate members that do not have the ability to synthesize PHA in PHA-MMCs, which would enhance the utilization efficiency of carbon sources. Second, the co-evolution of phages and PHA-synthesizing bacteria could be used to obtain strains with stronger antiviral capabilities. Third, phages carrying

AMGs related to PHA synthesis could be applied to enhance the PHA synthesis capacity of community members.

## MATERIALS AND METHODS

### PHA-MMCs enrichment

Five sequencing batch reactors (SBR), each with a working volume of 2 L, were used for the cultivation of PHA-MMCs. Activated sludge from a municipal wastewater treatment plant (Chengdu, China) was used as the inoculum source. Activated sludge was collected from the aerobic stage of the Anaerobic-Anoxic-Oxic process and filtered to remove solid impurities (screen size of 0.25 mm). Reactors A, B, C, D, and E were fed with synthetic wastewater containing acetate, propionate, butyrate, valerate, and lactate as the sole carbon source, respectively (Fig. 1A). The content of inorganic components and trace elements in synthetic wastewater is described in Supplemental information S1. Five SBRs were operated according to the ADF selection strategy, with a cycle length of 12 h, consisting of feeding (10 min), aeration (650 min), settling (50 min), and discharge (10 min). All reactors were operated at 26°C. More detailed information can be found in Supplemental information S1 and Fig. S23.

The maximum PHA accumulation content (wt%) of PHA-MMCs at different times was evaluated by fed-batch PHA accumulation assays. The dissolved oxygen (DO) concentration was maintained at above 4 mg/L by aeration. The DO meter (WTW, Germany) was used to detect the DO concentration. The 200 mL enriched sludge from each reactor was transferred to a new 400 mL reaction vessel, and then 200 mL of fresh substrate was added. Air was supplied to the reactor through an aeration pump. After aeration for 1 hour, aeration was stopped. After standing for 15 minutes, 200 mL of supernatant was removed. Subsequently, 200 mL of fresh substrate was added for the second batch of fermentation. The above operation was repeated for another three times. After the fifth fermentation, 10 mL of fermentation liquid was taken and centrifuged at 10,000 rpm for 10 min to collect the biomass. The biomass was pre-treated and the PHA content of biomass was determined using GC-MS. The detailed determination method was described in Supplemental information S2.

### Sample collection, DNA/RNA extraction, and sequencing

During the 145 days of the SBR operation, 10 time points were selected for biological sampling (Fig. 1B). By analyzing the abundance of dominant species in different reactors, we found that the dominant species in each reactor were in a state of constant fluctuation. To allow the meta-omics data to cover most of the dominant species in each reactor, we selected samples at different time points according to the species abundance curve (Fig. S24). The fermentation broth was collected in sterile DNase- and RNase-free centrifuge tubes and centrifuged at 12,000 rpm for 30 min at 4°C. The biomass was collected and stored in liquid nitrogen. Total DNA and RNA were extracted via the cetyl-trimethyl ammonium bromide (CTAB) method (60). DNA and RNA sequencing was performed on Illumina NovaSeq (Illumina Inc., San Diego, CA, USA) at Majorbio Bio-Pharm Technology Co., Ltd. (Shanghai, China). The size of each metagenomic and metatranscriptomic raw sequence data was more than 15 GB and 9 GB, respectively. Sequence data associated have been deposited in the NCBI Sequence Read Archive database (PRJNA1121485; the accession numbers can be found in Table S9). Details of library construction and sequencing are described in Supplemental information S3. The Trimmomatic v0.36 (61) was used to process raw DNA and RNA sequencing data to get clean raw data (ILLUMINACLIP: adapters/TruSeq3-PE.fa:2:30:10; LEADING:3; TRAILING:3; SLIDINGWINDOW:6:30; MINLEN:100).

## Bacterial MAGs assembly and bioinformatics analyses

The clean DNA reads from 30 metagenomes were spliced into contigs by SPAdes v.3.5.0 (62). The bacterial MAGs were assembled from contigs by MetaWRAP v.1.2.1 (63). MAGs with completeness ≥70% and contamination ≤10% were selected for subsequent analysis. The taxonomy of MAGs was classified by GTDB-Tk version 2.1.1 based on the Genome Taxonomy Database (64). The metagenomic and metatranscriptomic reads were mapped to MAGs by BBMap (v35.85; http://sourceforge.net/projects/bbmap/). The abundance and activity of MAGs were calculated as reads per kilobase transcript per million reads (RPKM).

$$\text{Abundance or activity of MAG} = \frac{\text{reads(MAG)}}{\frac{\text{size(MAG)}}{1,000}\frac{\text{reads(all MAGs)}}{100,000}}$$

In the equation, the "reads (MAG)" is the number of reads of metagenomic (or metatranscriptomic) raw data mapped to each MAG. The "reads (all MAGs)" is the sum of reads of all MAGs. The "size (MAG)" is the number of base pairs of each MAG. To compare differences in microbial communities at different enrichment times, all MAGs from the same reactor were clustered using dRep v.2.6.2 (-pa 0.9, -sa 0.99) (20). The abundance and activity values of clustered MAGs at different time points were summarized as abundance or activity (Tables S1 to S5) for subsequent statistical analysis. The CRISPR spacers were searched from MAG scaffolds by the CRISPR recognition tool (CRT, v2.1) (65). The antiphage systems of MAGs were detected by DefenseFinder v.1.2.2 (45).

## APC identification and bioinformatics analyses

The APCs were identified comprehensively from metagenomic contigs by VirSorter2 v.2.1.0 (28) and VirFinder v1.1 (29). A contig (length ≥5 kb) that met one of the following three conditions was selected as an APC: (i) VirSorter v2.1 score ≥0.9; (ii) VirFinder v1.1 score ≥0.9 along with $P < 0.05$; and (iii) both VirSorter2 score ≥0.5 and VirFinder ≥ 0.7, $P < 0.05$. CD-HIT v.4.7 was used to cluster the APCs to obtain pOTUs with parameter -c at 0.95 and -aS at 0.8. The abundance of pOTUs was calculated as reads per kilobase transcript per million reads (RPKM) by BBMap (v35.85; http://sourceforge.net/projects/bbmap/).

$$\text{Abundance of pOTU} = \frac{\text{reads(pOTU)}}{\frac{\text{size(pOTU)}}{1,000}\frac{\text{reads(all pOTUs)}}{100,000}}$$

In the equation, the "reads (pOTU)" is the number of reads of metagenomic raw data mapped to each pOTU. The "reads (all pOTUs)" is the sum of reads of all pOTUs. The "size(pOTU)" is the number of base pairs of each pOTU. The abundance of pOTUs in five SBRs is described in Tables S1 to S5. The CheckV v.1.0.3 (31) was used to assess the quality of pOTUs, and those with completeness ≥90% and contamination ≤60% were used for the subsequent analysis. The taxonomy of pOTUs was classified by PhaGCN2 v.2.0 (34). The lifestyle of pOTUs was predicted by the PhaTYP (33) function in PhaBOX (66).

The AMGs of pOTUs were identified and annotated by the DRAM-V v.1.5.0 (67) pipeline as follows: The VirSorter2 v.2.1.0 (28) (--prep-for-dramv) was run on pOTU sequences, and then the AMGs were predicted from the resulting sequences by DRAM-V v.1.5.0 (67). The putative AMGs were further confirmed using the NCBI CD-search tool (https://www.ncbi.nlm.nih.gov/Structure/cdd/wrpsb.cgi) with a threshold value of e-value $<10^{-5}$.

The prediction of phage-host linkages was performed on two different *in silico* strategies: (i) CRISPR-spacer match. The CRISPR spacers of MAGs were queried for exact matches to pOTUs sequences by BLASTn v.2.9.0 (68). Only the matches (identity ≥ 97%, coverage ≥ 90%, mismatch ≤ 1) were regarded as highly confident phage-host linkages. (ii) The sequence homology between MAGs and pOTUs. The sequences of pOTUs were compared with MAGs by BLASTn v.2.9.0 (68). The matches with identity ≥70%,

bit score ≥50, alignment length ≥2,500, and e-value ≤0.001 were regarded as highly confident phage-host linkages. Based on these potential phage-host linkages, the genus-level phage-host network was constructed and visualized by Gephi (0.10.1).

## Statistical analysis and visualization

The α diversity indexes and principal coordinates analysis (PCoA) of phage and microbial community were calculated in R v.4.3.2 with package "phyloseq" [69]. Based on the abundance table of community members, the R package "vegan" was used to calculate the Bray-Curtis dissimilarity between bacterial and phage communities at adjacent time points to characterize the degree of change in community member structure [42]. The Spearman correlation coefficient and $P$-value between the Bray-Curtis dissimilarity of bacterial and phage communities were provided by the function cor.test() in R v.4.3.2. The MRS [36] was used to quantitatively characterize this dynamic change of phage and bacterial communities. MRS is a temporal analog of species rank-abundance distributions and indicates the degree of species reordering between two time points. $\text{MRS} = \sum_{i=1}^{N} (|R_{i,t+1} - R_{i,t}|)/N$, where $N$ is the number of species in common in both time points, $t$ is the time point, and $R_{i,t}$ is the relative rank of species $i$ at time $t$. To further analyze the impact of phage communities on bacterial communities, SEM was constructed with phage community characteristics (α diversity, abundance, and activity) as predictor variables and bacterial cominmunity characteristics (α diversity, abundance, and activity) as response variables. The composition is represented by the PC1 from the Bray-Curtis dissimilarity-based principal coordinate analyses. The α diversity is represented by the Chao1 index. The phage abundance is the sum of the abundance value of each phage in the sample based on the metagenomic data. The bacterial activity is the sum of the activity value of each bacterial MAG in the sample based on the metatranscriptomic data. The SEM analysis was performed in R v.4.3.2 with the package "piecewiseSEM" v.2.3.0 (https://jslefche.github.io/piecewiseSEM/). The network diagram was visualized by Gephi v.0.10. The heatmap, phylogenetic tree, and gene cluster map were visualized by chiplot (https://www.chiplot.online/).

## ACKNOWLEDGMENTS

This study was funded by the China Petroleum and Chemical Corporation (No. 421063-10).

Y.-Q.T. and J.Y. conceived the project and designed the experiments. J.Y. did the experiments, analyzed the data, and wrote the manuscript. Y.-Q.T. supervised the project and revised the manuscript. Y.Z., Y.C., M.G., and Z.-Y.X. contributed to the methodology. X.H., M.W., and Q.Z. contributed to sample collection and nucleic acid extraction.

## AUTHOR AFFILIATIONS

[1]College of Architecture and Environment, , Sichuan University, Chengdu, Sichuan, China
[2]Sichuan Environmental Protection Key Laboratory of Organic Wastes Valorization, Chengdu, Sichuan, China
[3]Sinopec Shanghai Engineering Co. Ltd., Shanghai, China
[4]Sinopec (Dalian) Research Institute of Petroleum and Petrochemicals Co. Ltd., Dalian, Liaoning, China
[5]Institute for Disaster Management and Reconstruction, Sichuan University, Chengdu, Sichuan, China
[6]Engineering Research Centre of Alternative Energy Materials and Devices, Ministry of Education, Chengdu, Sichuan, China

## AUTHOR ORCIDs

Jian Yao ⓘ http://orcid.org/0000-0003-3121-4223

Yue-Qin Tang http://orcid.org/0000-0001-6872-1099

## FUNDING

| Funder | Grant(s) | Author(s) |
|---|---|---|
| China Petrochemical Corporation | No. 421063-10 | Yan Zeng |
| | | Xia Hong |
| | | Meng Wang |
| | | Quan Zhang |
| | | Yating Chen |
| | | Min Gou |
| | | Zi-Yuan Xia |
| | | Yue-Qin Tang |

## AUTHOR CONTRIBUTIONS

Jian Yao, Data curation, Formal analysis, Methodology, Writing – original draft | Yan Zeng, Methodology | Xia Hong, Funding acquisition | Meng Wang, Funding acquisition | Quan Zhang, Funding acquisition | Yating Chen, Methodology | Min Gou, Methodology | Zi-Yuan Xia, Methodology | Yue-Qin Tang, Funding acquisition, Supervision, Writing – review and editing

## DATA AVAILABILITY

Sequence data associated have been deposited in the NCBI Sequence Read Archive database (PRJNA1121485).

## ETHICS APPROVAL

No animals or humans were involved in this study.

## ADDITIONAL FILES

The following material is available online.

### Supplemental Material

**Supplemental material (mSystems00200-25-s0001.pdf).** Supplemental figures and information.
**Table S1 (mSystems00200-25-s0002.xlsx).** The abundance (activity) of MAGs and the abundance of pOTUs in Reactor A.
**Table S2 (mSystems00200-25-s0003.xlsx).** The abundance (activity) of MAGs and the abundance of pOTUs in Reactor B.
**Table S3 (mSystems00200-25-s0004.xlsx).** The abundance (activity) of MAGs and the abundance of pOTUs in Reactor C.
**Table S4 (mSystems00200-25-s0005.xlsx).** The abundance (activity) of MAGs and the abundance of pOTUs in Reactor D.
**Table S5 (mSystems00200-25-s0006.xlsx).** The abundance (activity) of MAGs and the abundance of pOTUs in Reactor E.
**Table S6 (mSystems00200-25-s0007.xlsx).** The statistical characteristics of the interaction network.
**Table S7 (mSystems00200-25-s0008.xlsx).** The sequence similarity among phages of each host.
**Table S8 (mSystems00200-25-s0009.xlsx).** The activity of AMGs in MAGs and pOTUs.
**Table S9 and S10 (mSystems00200-25-s0010.xlsx).** Accession numbers of sequence data and sequences of AMGs.

Open Peer Review

**PEER REVIEW HISTORY (review-history.pdf).** An accounting of the reviewer comments and feedback.

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
