## [Reviewer comments · mSystems]

Phages-bacteria interactions underlying the dynamics of polyhydroxyalkanoates-producing mixed microbial cultures via meta-omics study

Jian Yao, Yan Zeng, Xia Hong, Meng Wang, Quan Zhang, Ya-Ting Chen, Min Gou, Zi-Yuan Xia, and Yue-Qin Tang

Corresponding Author(s): Yue-Qin Tang, Sichuan University

Review Timeline:

Submission Date:	February 10, 2025
Editorial Decision:	February 12, 2025
Revision Received:	February 19, 2025
Accepted:	February 26, 2025

Editor: Liyuan Ma

Reviewer(s): The reviewers have opted to remain anonymous.

Transaction Report:

DOI: <https://doi.org/10.1128/msystems.00200-25>

Re: mSystems00200-25 (Phages-bacteria interactions underlying the dynamics of polyhydroxyalkanoates-producing mixed microbial cultures via meta-omics study)

Dear Prof. Yue-Qin Tang:

Thank you for your resubmission. Although the decision on the full-text evaluation is Minor Modifications this time, it still needs to be carefully or even Major Revised according to the reviewer's comments. Hypotheses should be given. Innovation of the manuscript (compared to <https://doi.org/10.1016/j.cub.2023.06.033>)?

Please return the manuscript within 30 days; if you cannot complete the modification within this time period, please contact me. If you do not wish to modify the manuscript and prefer to submit it to another journal, notify me immediately so that the manuscript may be formally withdrawn from consideration by mSystems.

Revision Guidelines

Sincerely,
Liyuan Ma
Editor
mSystems

Reviewer #3 (Comments for the Author):

Thank you for the rapid revision!

Hypothesis is still missing.

R1: Well, they have three replicates within each treatment category, namely, "established wells" and "new wells". In your case, the treatments were created by the addition of different carbon source but lacked any replicates within each treatment category. A proper experimental design should consist of at least three replicates. Therefore, your response in R22 is invalid because your reactors were not the same environment, you definitely created distinct conditions by applying different carbon sources.

R2: I understand the difficulty here but your speculative sentences have not been resolved. You could also mention this as a potential future study idea.

Figure 2.: Absolute or relative abundance? Needs to be clarified.

R3: No, it is still there in figures.

R4: Your sentence has a slightly different meaning. Your results indicate that the EXTENT OF SHIFTS were greatly similar. In L299, you wrote about host survival but, for me, it is more indicative for phage specificity, meaning that phage composition changes because the same dominant phage cannot adapt to a newly emerged dominant bacterial taxon. So they are quite picky and cannot broadly target hosts. Note also that, although Bray-Curtis distances carry some abundance information, it is more mirroring compositional changes/dissimilarities between time points (in your case). If phages can target multiple hosts, why is there a synchronous change with bacterial compositional shifts? To also emphasize my second question here again, why is there an increase of PHA content then? What is actually causing the enrichment of PHA content?

Figure 2: Why do you have coloured heatmap for only bacteria? In case of phage contig lengths, could you detect dissimilarities between reactors?

Figure 3C: As Bray-Curtis dissimilarity cannot have a value of more than 1, I recommend to set a limit for the y-axis.

R18: It is quite impossible to assess the impact of something if there is no control (reference) treatment. One can easily find articles to any scenario but published research are never the truth set in stone! We should always question anyone's findings and approaches and critically evaluate each others' works.

R19: Yes but your results suggest that each dominant groups get eradicated sooner or later (you speculate that this is caused by phages rather than other community assembly mechanisms) which altogether suggests the presence of lytic phages, even though you detect more or less the same amount of lysogenic phages. So, the whole story is a bit puzzling and based on speculation when it comes to conclusions about phages' role in PHA synthesis in this study. Nonetheless, I see your point and it could be the truth.

R20: It is still here: L278-284, carrying the same message.

R25: Figure 3: Please correct the typo: "comunity". Also, it would be great if you could use the same reactor-indicating colours in i.e., Fig 2A, to be consistent.

R28: Okay, but I highly recommend to check other methods, like Mantel or Procrustes tests in the future. They are more suitable for distance matrices.

R29: That is not so good. I guess here you used the `cor.test()` values as input. Apply at least $r=0.6$ values (or even higher ones). This is a common approach when it comes to network construction. But spurious interactions should be eliminated by using more suitable and reliable methods, e.g., FlashWeave that can deal with the problem of compositionality and taking into accounts co-distribution merely by sharing niche optima. So all in all, this network analysis should be seen and interpreted with a grain of salt...

L41-43: Affected in what way? Needs to be specified!

L600: Phage activity is not presented in the SEM result. What is phage activity? How did you measure it? And how bacterial activity is measured? It is unclear! Needs to be specified in the methods, as well as at the corresponding results in the main text.

Conclusion is nicely written!

Dear Editor Liyuan Ma

On behalf of my co-authors, we thank you very much for giving us another opportunity to revise our manuscript. We appreciate editor and reviewers very much for their positive and constructive comments and suggestions on our manuscript entitled “Phages-bacteria interactions underlying the dynamics of polyhydroxyalkanoates-producing mixed microbial cultures via meta-omics study” (submission ID mSystems00200-25).

We have studied the Reviewers’ comments carefully and have made revision which marked in tracked changes in the revised submission file. We have tried our best to revise our manuscript according to the comments and we have addressed the hypothesis development and taken this opportunity to re-emphasize the key innovations of our work throughout the manuscript. Attached please find the revised version, which we would like to submit for your kind consideration.

We would like to express our great appreciation to you and reviewers for comments on our paper. Looking forward to hearing from you.

Thank you and best regards.

Yours sincerely,

Corresponding author

Yue-Qin Tang

Dear Reviewer3

On behalf of my co-authors, thank you very much for your comments and professional advices. These opinions help to improve academic rigor of our article. Based on your suggestion and request, we have made corrections to the revised manuscript. We have carefully considered your concerns and incorporated corresponding discussions in the manuscript. We hope these modifications align with your expectations. We would like to show the details as follows:

Q1: Hypothesis is still missing.

R1: We are sorry for this weak point. We have adjusted the introduction section to highlight our hypothesis.

Before revision:

Currently, the process for the production of PHAs by MMCs using complex waste biomass has three main steps (5). The process includes: 1) acid production: producing short-chain carboxylic acids (SCCAs, such as acetate, propionate, butyrate, valerate and lactate) that serve as substrates of PHAs production by anaerobic fermentation; 2) MMCs enrichment: enriching microbial communities with PHAs-synthesizing capacity from activated sludge by applying selective pressure; 3) PHAs accumulation: the maximum PHAs accumulation of enriched MMCs using SCCAs. Of these, the second step is the most important in the process. Nowadays, aerobic dynamic feeding (ADF) is the main strategy to enrich PHA-MMCs (6). Alternating cycles of the feast (carbon rich) and famine (carbon poor) in the ADF provides selection pressure. Because PHA producers can accumulate intracellular PHAs as a backup carbon source during the feast phase, they are better able to gain a competitive advantage during the famine phase (7). However, the dominant species in enriched microbial communities continue to change even under stable maintenance conditions (7). Huang et al. suggest that this is due to the "kill the winner" effect of phages where virulent phages prey on dominant species, causing other species to gain a dominant position and resulting in repeated replacement of dominant species(7). Eriksen et al. show that

the phages in soil can drive the succession of bacterial communities by “killing the winner”(8). To date, the effect of phages on microbial community assembly during the enrichment and maintenance of PHA-MMCs has not been studied.

In recent years, researchers have begun to uncover the effects of phages on the structure and function of microbial communities in natural environments (e.g., deep-sea sediment (9) and soil (10)) and engineered systems (e.g., municipal wastewater treatment plants (11-13)). Virulent phages hijack the hosts to produce progeny and eventually lyse the host, and host cell debris fuel other microorganisms, which can directly affect the structure of microbial communities (14, 15). In addition, both virulent and temperate phages can manipulate host metabolism by reprogramming host metabolic pathways or expressing auxiliary metabolic genes (AMGs) to influence the function of microbial communities. For example, in the ocean, many AMGs are involved in photosynthesis (16), the pentose phosphate pathway (17), nitrogen metabolism (18), and sulfur metabolism (19). Given that the enrichment of PHA-MMCs is aimed at increasing PHA synthesis ability, phages may play an important but yet undiscovered role by influencing the abundance of dominant species or encoding AMGs related to PHA synthesis. Therefore, studies are needed to examine the phage-host relationship in PHA-MMCs and extra PHA metabolic capabilities conferred by AMGs in phages.

After revision:

Currently, the process for the production of PHAs by MMCs using complex waste biomass has three main steps (5). The process includes: 1) acid production: producing short-chain carboxylic acids (SCCAs, such as acetate, propionate, butyrate, valerate and lactate) that serve as substrates of PHAs production by anaerobic fermentation; 2) MMCs enrichment: enriching microbial communities with PHAs-synthesizing capacity from activated sludge by applying selective pressure; 3) PHAs accumulation: the maximum PHAs accumulation of enriched MMCs using SCCAs. Of these, the second step is the most important in the process. Nowadays, aerobic dynamic feeding

(ADF) is the main strategy to enrich PHA-MMCs (6). Alternating cycles of the feast (carbon rich) and famine (carbon poor) in the ADF provide selection pressure. Because PHA producers can accumulate intracellular PHAs as a backup carbon source during the feast phase, they are better able to gain a competitive advantage during the famine phase (7). However, the dominant species in enriched microbial communities continue to change even under stable maintenance conditions and this species turnover cannot be fully explained by interspecific competition(7). (Line 99-101)

In recent years, researchers have begun to uncover the effects of phages on the structure and function of microbial communities in natural environments (e.g., deep-sea sediment (8) and soil (9)) and engineered systems (e.g., municipal wastewater treatment plants (10-12)). Virulent phages hijack hosts to produce progeny and eventually lyse hosts, and host cells debris fuel other microorganisms, which can directly affect the structure of microbial communities (13, 14). In addition, both virulent and temperate phages can manipulate host metabolism by reprogramming host metabolic pathways or expressing auxiliary metabolic genes (AMGs) to influence the function of microbial communities. For example, in the ocean, many AMGs are involved in photosynthesis (15), the pentose phosphate pathway (16), nitrogen metabolism (17), and sulfur metabolism (18). Given that PHA-MMCs are cultivated in open systems, the bacterial communities within may also be influenced by phages. In light of the above considerations, we hypothesize that phages constitute a factor affecting the structure of bacterial communities in PHA-MMCs and may carry auxiliary metabolic genes (AMGs) associated with PHA synthesis. (Line 112-116)

Q2: R1: Well, they have three replicates within each treatment category, namely, "established wells" and "new wells". In your case, the treatments were created by the addition of different carbon source but lacked any replicates within each treatment category. A proper experimental design should consist of at least three replicates. Therefore, your response in R22 is invalid because your reactors were not the same environment, you definitely created distinct conditions by applying different carbon sources.

R2: Your perspective is entirely valid and rigorous. Our study aimed to investigate the impact of phages on bacterial community structure across PHA-MMCs enriched under varying carbon sources. The strategy we used was grounded in the premise that cross-validation among PHA-MMCs systems enriched with distinct carbon sources could reveal more generalizable patterns of phage-mediated community dynamics. However, the limited replication within individual experimental systems represents a methodological constraint that compromises the robustness of our conclusions. We have explicitly acknowledged this limitation in the manuscript and intend to refine future experimental designs by incorporating enhanced replication protocols, as you suggested. We discussed it carefully in the manuscript, as follows:

“In this study, we investigated the phage communities within PHA-MMCs and their potential impacts on bacterial community structure and function, though several methodological limitations remain, which we aim to address in future research. First, while cross-system comparisons among five PHA-MMCs enriched with distinct carbon sources were intended to identify generalized patterns of phage-mediated bacterial community dynamics, setting parallel reactor systems would further prove these findings. Subsequent studies should consider reactor replication and systematic validation across different systems to rigorously elucidate the role of phages in shaping bacterial communities.” (Line 444-452)

Q3: R2: I understand the difficulty here but your speculative sentences have not been resolved. You could also mention this as a potential future study idea.

R3: We appreciate your patient guidance. We have discussed this in detail in the manuscript, as follows:

“The third limitation concerns the need for further investigation into the influence of phage-encoded AMGs on bacterial community-mediated PHA synthesis. The finding that some phages within PHA-MMCs harbored transcriptionally active AMGs suggests that these phages may provide a critical genetic reservoir for bacterial members. However, the potential contribution of phage-carried AMGs on the PHA synthesis capacity of bacterial communities requires controlled experimental validation. Furthermore, whether lytic and lysogenic phages carrying AMGs exert equivalent impacts on the PHA synthesis efficiency of bacterial communities remains unresolved. Notably, excessive suppression of hosts by lytic phages could negate any potential enhancement of overall PHA yield. In subsequent studies, more rigorous controlled trials need to be conducted. It is needed to isolate lytic and lysogenic phages harboring AMGs, scale up their cultivation, supply them into PHA-MMCs, and systematically monitor changes in PHA synthesis efficiency of bacterial communities.” (Line 462-474)

Q4: Figure 2.: Absolute or relative abundance? Needs to be clarified.

R4: We are sorry for this mistake. We have modified Figure 2.

Before revision:

Figure 2 The overview of bacterial and phage communities

(A) The maximum PHA accumulation content (wt%) of PHA-MMCs in each reactor. (B) The top 10 bacterial genera in each reactor. (C) Distribution of pOTU sequence lengths. (D) The percentage of pOTUs of different quality. (E) The top 10 pOTU genera in each reactor.

After revision:

Figure 2 The overview of bacterial and phage communities

(A) The maximum PHA accumulation content (wt%) of PHA-MMCs in each reactor. (B) The top 10 bacterial genera (relative abundance) in each reactor. (C) Distribution of pOTU sequence lengths. (D) The percentage of pOTUs of different quality. (E) The top 10 pOTU genera (relative abundance) in each reactor.

Q5: R3: No, it is still there in figures.

R5: We are very sorry for this mistake. We have revised it in Figure 3D

Before revision:

After revision:

Q6: R4: Your sentence has a slightly different meaning. Your results indicate that the EXTENT OF SHIFTS were greatly similar. In L299, you wrote about host survival but, for me, it is more indicative for phage specificity, meaning that phage composition changes because the same dominant phage cannot adapt to a newly emerged dominant bacterial taxon. So they are quite picky and cannot broadly target hosts. Note also that, although Bray-Curtis distances carry some abundance information, it is more mirroring compositional changes/dissimilarities between time points (in your case). If phages can target multiple hosts, why is there a synchronous change with bacterial compositional shifts? To also emphasize my second question here again, why is there an increase of PHA content then? What is actually causing the enrichment of PHA content?

R6: This is a highly significant and thought-provoking question. In our study, a single phage could indeed establish associations with multiple hosts, as evidenced by CRISPR-spacer matches or sequence homology between metagenome-assembled genomes (MAGs) and phage operational taxonomic units (pOTUs). Whether such host diversity contributes to the stabilization of phage abundance and influences the coherence of structural dynamics between phages and bacterial communities remains an open question requiring further investigation. We have discussed this issue in the manuscript as follows:

“The second limitation pertains to the prolonged intervals between sampling points, which compromised the ability to resolve more nuanced synchronous dynamics between phage and host abundance. Approximately half of the phages detected in this study exhibited multiple potential hosts, raising the question of whether such host generality may contribute to phage abundance and thereby obscure the coherence of phage and bacterial community structural shifts. Future studies should implement high-frequency sampling protocols within each of PHA-MMCs systems to rigorously

track temporal fluctuations in phage and host abundance and concurrent community structural adaptations.” (Line 453-461)

Regarding the variations in PHA content across individual reactors, the increase in PHA levels during the enrichment phase can be attributed to the gradual dominance of PHA-synthesizing bacteria in the community under the ADF enrichment strategy. During the maintenance phase, PHA content remained stable in all reactors. To avoid potential misinterpretations, we have revised the manuscript to clarify the description of the impact of AMGs on bacterial PHA synthesis capacity and have included a detailed discussion of this aspect in the manuscript.

Revision in the Importance section:

Before revision:

Phage communities can regulate the activity and structure of bacterial communities. In addition, the AMGs related to PHA synthesis carried by phages might promote the PHA synthesis ability of bacterial members.

After revision:

Phage communities can regulate the activity and structure of bacterial communities. In addition, AMGs related to PHA synthesis may hitchhike during phage-host infection cycles, enabling their dissemination across bacterial communities, and phages may act as a critical genetic reservoir for bacterial members, facilitating access to PHA synthesis-related functional traits. (Line 58-63)

Revision in the Discussion section:

Before revision:

However, these AMGs carried by phages were transcriptionally active in different samples (Table S8). This suggests a method to enhance the PHA synthesis capacity of PHA-MMCs in situ. Temperate phages carrying AMGs related to PHA anabolism could be isolated and cultured, and then added back to the PHA-MMCs. The expression of these AMGs would possibly enhance the activity of the PHA synthesis pathway in PHA-MMCs.

After revision:

However, these AMGs carried by phages were transcriptionally active in different samples (Table S8). These findings suggest that AMGs associated with PHA synthesis may hitchhike during phage-host infection cycles, enabling their dissemination across bacterial communities. Furthermore, phages may act as a critical genetic reservoir for bacterial members, facilitating access to PHA synthesis-related functional traits. (Line 434-439)

Added discussion:

“The third limitation concerns the need for further investigation into the influence of phage-encoded AMGs on bacterial community-mediated PHA synthesis. The finding that some phages within PHA-MMCs harbored transcriptionally active AMGs suggests that these phages may provide a critical genetic reservoir for bacterial members. However, the potential contribution of phage-carried AMGs on the PHA synthesis capacity of bacterial communities requires controlled experimental validation. Furthermore, whether lytic and lysogenic phages carrying AMGs exert equivalent impacts on the PHA synthesis efficiency of bacterial communities remains unresolved. Notably, excessive suppression of hosts by lytic phages could negate any potential enhancement of overall PHA yield. In subsequent studies, more rigorous controlled trials need to be conducted. It is needed to isolate lytic and lysogenic phages harboring AMGs, scale up their cultivation, supply them into PHA-MMCs,

and systematically monitor changes in PHA synthesis efficiency of bacterial communities.” (Line 462-474)

Q7: Figure 2: Why do you have coloured heatmap for only bacteria? In case of phage contig lengths, could you detect dissimilarities between reactors?

R7: I have adjusted the colours of both heatmaps to match.

We plotted boxplots based on the length of phage sequences assembled in different reactors and performed one-way ANOVA analysis. The results showed that there was no significant difference in the length distribution of phage sequences among different reactors.

Before revision:

After revision:

The length of phage contigs in each reactor. The “ns” represents “no significant difference”

Q8: Figure 3C: As Bray-Curtis dissimilarity cannot have a value of more than 1, I recommend to set a limit for the y-axis.

R8: We are very grateful for this important reminder. We have made changes to Figure 3C.

Before revision:

After revision:

Q9: R18: It is quite impossible to assess the impact of something if there is no control (reference) treatment. One can easily find articles to any scenario but published research are never the truth set in stone! We should always question anyone's findings and approaches and critically evaluate each others' works.

R9: Your perspective is decisively rigorous and scientific, which also suggests that we

should correctly view the work of others and how to guide our own research endeavors. Consequently, in our subsequent studies, we will implement stringent controlled experiments to verify the impact of auxiliary metabolic genes on the PHA synthesis capability of bacterial members. The importance of controlled trials has been underscored in our draft as follows:

“In subsequent studies, more rigorous controlled trials need to be conducted. It is needed to isolate lytic and lysogenic phages harboring AMG, scale up their cultivation, supply them into PHA-MMCs, and systematically monitor changes in PHA synthesis efficiency of bacterial communities.

” (Line 471-474)

Q10: R19: Yes but your results suggest that each dominant groups get eradicated sooner or later (you speculate that this is caused by phages rather than other community assembly mechanisms) which altogether suggests the presence of lytic phages, even though you detect more or less the same amount of lysogenic phages. So, the whole story is a bit puzzling and based on speculation when it comes to conclusions about phages' role in PHA synthesis in this study. Nonetheless, I see your point and it could be the truth.

R10: This represents a critical scientific issue worthy of in-depth investigation: whether virulent phages and lysogenic phages carrying AMG exert significantly different impacts on PHA biosynthesis in bacterial hosts. Accordingly, we have included dedicated discussion in the manuscript as follows:

“Furthermore, whether lytic and lysogenic phages carrying AMG exert equivalent impacts on the PHA synthesis efficiency of bacterial communities remains unresolved. Notably, excessive suppression of hosts by lytic phages could negate any potential enhancement of overall PHA yield. In subsequent studies, more rigorous controlled trials need to be conducted. It is needed to isolate lytic and lysogenic phages harboring AMG, scale up their cultivation, supply them into PHA-MMCs, and systematically monitor changes in PHA synthesis efficiency of bacterial communities.”

(Line 468-474)

Q11: R20: It is still here: L278-284, carrying the same message.

R11: We apologize for any lack of clarity in this explanation.

Based on our 16S rRNA sequencing results, the Chao1 index of bacterial communities in each reactor remained dynamic and fluctuating throughout the extended operational period. This may suggest that virulent phages suppress host populations without driving host extinction, or alternatively, reflect the resilience inherent to complex microbial consortia. Moreover, the open-system configuration of these five reactors permitted potential colonization by exogenous bacterial populations under favorable conditions. To prevent potential misinterpretation, we have removed the contentious statements regarding this phenomenon from the revised manuscript.

Before revision:

Importantly, there was a significant negative correlation ($r: -0.43, p < 0.05$) between phage abundance and the alpha diversity (Chao1) index of the bacterial community, in addition to a negative correlation ($r: -0.36$) with bacterial community activity. This might indicate that virulent phages hijacked and killed host cells during proliferation, thereby reducing the activity of the bacterial hosts. ~~As the phage proliferated and lysed host cells, the number of hosts rapidly decreased, which in turn reduced the Chao1 index of the bacterial community.~~ This further suggested that the synergistic changes between the phage and bacterial communities were not caused by the

response of the temperate phages to the dynamics of hosts, as they benefit from the high growth rate of the host while limiting their lytic activity (37, 39).

After revision:

Importantly, there was a significant negative correlation ($r: -0.43, p < 0.05$) between phage abundance and the alpha diversity (Chao1) index of the bacterial community, in addition to a negative correlation ($r: -0.36$) with bacterial community activity. This might indicate that virulent phages hijacked and killed host cells during proliferation, thereby reducing the activity of the bacterial hosts. This further suggested that the synergistic changes between the phage and bacterial communities were not caused by the response of the temperate phages to the dynamics of hosts, as they benefit from the high growth rate of the host while limiting their lytic activity (37, 39). (Line 264-272)

Q12: R25: Figure 3: Please correct the typo: "comunity". Also, it would be great if you could use the same reactor-indicating colours in i.e., Fig 2A, to be consistent.

R12: We are very sorry for this mistake. We have corrected and adjusted the colours in Figures 3A&B.

Before revision:

After revision:

Q13: R28: Okay, but I highly recommend to check other methods, like Mantel or Procrustes tests in the future. They are more suitable for distance matrices.

R13: We are very grateful for your completely correct suggestions. We will choose the appropriate statistical methods based on your suggestions in our subsequent research.

Q14: R29: That is not so good. I guess here you used the cor.test() values as input. Apply at least $r=0.6$ values (or even higher ones). This is a common approach when it comes to network construction. But spurious interactions should be eliminated by using more suitable and reliable methods, e.g., FlashWeave that can deal with the problem of compositionality and taking into accounts co-distribution merely by sharing niche optima. So all in all, this network analysis should be seen and interpreted with a grain of salt...

R14: We sincerely apologize for this misunderstanding.

We would like to clarify that the identification of potential phage-host interactions was determined through CRISPR-spacer matches and sequence homology between

MAGs and pOTUs. Specifically, a bacterial MAG was considered a potential host for a phage only when either of the following criteria was met: (1) reliable sequence similarity between CRISPR-spacers in bacterial MAGs and phage sequences, or (2) reliable sequence homology between phage sequences and bacterial MAGs. Based on this rigorous approach, we identified potential phage-host relationships and subsequently constructed the genus-level phage-host network (Figure 4A). This network was not constructed based on `cor.test()` results. To prevent further confusion, we have explicitly detailed the network construction methodology in the Materials and Methods section as follows:

“The prediction of phage-host linkages was performed on two different *in silico* strategies: (i) CRISPR-spacers match. The CRISPR spacers of MAGs were queried for exact matches to pOTUs sequences by BLASTn v.2.9.0 (68). Only the matches (identity $\geq 97\%$, coverage $\geq 90\%$, mismatch ≤ 1) were regarded as highly confident phage-host linkages. (ii) The sequence homology between MAGs and pOTUs. The sequences of pOTUs were compared with MAGs by BLASTn v.2.9.0 (68). The matches with identity $\geq 70\%$, bit score ≥ 50 , alignment length ≥ 2500 , and e-value ≤ 0.001 were regarded as highly confident phage-host linkages. Based on these potential phage-host linkages, the genus-level phage-host network was constructed and visualized by Gephi (0.10.1).” (Line 604-612)

Q15: L41-43: Affected in what way? Needs to be specified!

R15: We apologize for this unclear statement. We have made revisions in the manuscript, as follows:

Before revision:

The dynamic changes in the structure of phage and bacterial communities were remarkably consistent. Structural equation modeling analysis showed that phage communities significantly affected the activity and community structure of bacterial microorganisms.

After revision:

The dynamic changes in the structure of phage and bacterial communities were remarkably consistent. Structural equation modeling analysis showed that phages could infect and lyse dominant species to vacate ecological niches for other species, resulting in a community succession state in which dominant species alternated. (Line 39-42)

Q16: L600: Phage activity is not presented in the SEM result. What is phage activity? How did you measure it? And how bacterial activity is measured? It is unclear! Needs to be specified in the methods, as well as at the corresponding results in the main text.

R16: We apologize for the lack of clear definitions of phage abundance as well as bacterial activity in SEM.

In SEM analysis, phage abundance or activity is the sum of the abundance or activity value of each phage in the sample based on the metagenome or metatranscriptome data, and the abundance or activity for each phage is calculated as follows:

$$\text{abundance or activity of } pOTU = \frac{\text{reads}(pOTU)}{\frac{\text{Size}(pOTU)}{1000} \frac{\text{reads}(\text{all } pOTUs)}{100000}}$$

In the equation, the “reads (pOTU)” is the number of reads of metagenomic raw data mapped to each pOTU. The “reads (all pOTUs)” is the sum of reads of all pOTUs. The “size (pOTU)” is the number of base pairs of each pOTU.

Bacterial abundance or activity is the sum of the abundance or activity value of each bacterial MAG in the sample based on the metagenome or metatranscriptome data,

and the abundance or activity for each bacterial MAG is calculated as follows:

$$\text{abundance or activity of MAG} = \frac{\text{reads(MAG)}}{\frac{\text{Size(MAG)}}{1000} \frac{\text{reads(all MAGs)}}{100000}}$$

In the equation, the “reads (MAG)” is the number of reads of metagenomic (or metatranscriptomic) raw data mapped to each MAG. The “reads (all MAGs)” is the sum of reads of all MAGs. The “size (MAG)” is the number of base pairs of each MAG.

We contend that accurate assessment of phage activity presents significant challenges when relying solely on metatranscriptomic data. Furthermore, bacterial abundance metrics may not reliably reflect suppression by virulent phages, as these measurements inherently include both viable and non-viable bacterial cells. Consequently, in our structural equation modeling (SEM) analysis, we have specifically focused on examining the relationship between phage abundance and bacterial activity as more robust indicators of phage-host dynamics.

We have also added the definitions of phage abundance and bacterial activity to the caption of Figure 3D and the Materials and Methods section, as follows:

Revision in the Figure 3D caption:

“(D) Path diagram of SEM showing the effect of phage communities on bacterial communities. Composition is represented by the PC1 from the Bray-Curtis dissimilarity-based principal coordinate analyses. The α diversity is represented by the Chao1 index. The phage abundance is the sum of the abundance value of each phage in the sample based on the metagenomic data. The bacterial activity is the sum of the activity value of each bacterial metagenome assemble genome (MAG) in the sample based on the metatranscriptomic data.” (Line 882-886)

Revision in the Methods:

“To further analyze the impact of phage communities on bacterial communities, structural equation modeling (SEM) was constructed with phage community

characteristics (α diversity, abundance, and activity) as predictor variables and bacterial community characteristics (α diversity, abundance, and activity) as response variables. The composition is represented by the PC1 from the Bray-Curtis dissimilarity-based principal coordinate analyses. The α diversity is represented by the Chao1 index. The phage abundance is the sum of the abundance value of each phage in the sample based on the metagenomic data. The bacterial activity is the sum of the activity value of each bacterial MAG in the sample based on the metatranscriptomic data.” (Line 627-636)

Finally, we would like to express our sincere admiration for your expertise and rigorous approach to research. Your insightful comments have significantly contributed to the improvement of our work. Looking forward to hearing from you.

Thank you and best regards.

Yours sincerely,

Corresponding author

Yue-Qin Tang

Re: mSystems00200-25R1 (Phages-bacteria interactions underlying the dynamics of polyhydroxyalkanoates-producing mixed microbial cultures via meta-omics study)

Dear Prof. Yue-Qin Tang:

The authors have addressed all the comments successfully. I think the present version is ready to go. Congratulations.

Your manuscript has been accepted, and I am forwarding it to the ASM production staff for publication. Your paper will first be checked to make sure all elements meet the technical requirements. ASM staff will contact you if anything needs to be revised before copyediting and production can begin. Otherwise, you will be notified when your proofs are ready to be viewed.

Cover Image Submissions: If you would like to submit a potential Cover Image, please email a file and a short legend to mSystems@asmusa.org. Please note that we can only consider images that (i) the authors created or own and (ii) have not been previously published. By submitting, you agree that the image can be used under the same terms as the published article. Image File requirements: TIF/EPS, 7.5 inches wide by 8.25 inches tall (at least 2,250 pixels wide by 2,475 pixels tall), minimum 300 dpi resolution (600 dpi preferred), RGB, and no figure elements, e.g., arrows or panel labels. The legend should be a short description of the image, 1-2 sentences recommended. Please download and use this interactive template in Adobe to ensure that your proposed cover image meets our size requirements (<https://journals.asm.org/pb-assets/pdf-text-excel-files/ASM-Interactive-Sizing-Cover-Template-1715689791.pdf>).

Sincerely,
Liyuan Ma
Editor
mSystems